# The Effect of the COVID-19 Pandemic on the Social Inequalities of Health Care Use in Hungary: A Nationally Representative Cross-Sectional Study

**DOI:** 10.3390/ijerph19042258

**Published:** 2022-02-16

**Authors:** Bayu Begashaw Bekele, Bahaa Aldin Alhaffar, Rahul Naresh Wasnik, János Sándor

**Affiliations:** 1Doctoral School of Health Sciences, University of Debrecen, 4028 Debrecen, Hungary; baybeg121@gmail.com (B.B.B.); rahul.naresh@med.unideb.hu (R.N.W.); 2Department of Public Health, College of Health Sciences, Mizan Tepi University, Mizan Aman 260, Ethiopia; 3Department of Public Health and Epidemiology, Faculty of Medicine, University of Debrecen, 4028 Debrecen, Hungary; 4Faculty of Public Health, University of Debrecen, 4028 Debrecen, Hungary; bhaa.alhafar@gmail.com

**Keywords:** CRPNR, COVID-19 pandemic, hospital admission, GP visit, Hungary, interaction effect, Roma, specialist care

## Abstract

Background: The social representation of restricted health care use during the COVID-19 pandemic has not been evaluated properly yet in Hungary. Objective: Our study aimed to quantify the effect of COVID-19 pandemic measures on general practitioner (GP) visits, specialist care, hospitalization, and cost-related prescription nonredemption (CRPNR) among adults, and to identify the social strata susceptible to the pandemic effect. Methods: This cross-sectional study was based on nationally representative data of 6611 (N_prepandemic_ = 5603 and N_pandemic_ = 1008) adults. Multivariable logistic regression models were applied to determine the sociodemographic and clinical factors influencing health care use by odds ratios (ORs) along with the corresponding 95% confidence intervals (CI). To identify the social strata susceptible to the pandemic effect, the interaction of the time of data collection with the level of education, marital status, and Roma ethnicity, was tested and described by iORs. Results: While the CRPNR did not change, the frequency of GP visits, specialist care, and hospitalization rates was remarkably reduced by 22.2%, 26.4%, and 6.7%, respectively, during the pandemic. Roma proved to be not specifically affected by the pandemic in any studied aspect, and the pandemic restructuring of health care impacted the social subgroups evenly with respect to hospital care. However, the pandemic effect was weaker among primary educated adults (iOR_GP visits, high-school vs. primary-education_ = 0.434; 95% CI 0.243–0.776, OR_specialist visit, high-school vs. primary-education_ = 0.598; 95% CI 0.364–0.985), and stronger among married adults (iOR_GP visit, widowed vs. married_ = 2.284; 95% CI 1.043–4.998, iOR_specialist visit, widowed vs. married_ = 1.915; 95% CI 1.157–3.168), on the frequency of GP visits and specialist visits. The prepandemic CRPNR inequality by the level of education was increased (iOR_high-school vs. primary-education_ = 0.236; 95% CI 0.075–0.743). Conclusion: Primary educated and widowed adults did not follow the general trend, and their prepandemic health care use was not reduced during the pandemic. This shows that although the management of pandemic health care use restrictions was implemented by not increasing social inequity, the drug availability for primary educated individuals could require more support.

## 1. Introduction

The Coronavirus Disease (COVID-19) outbreak was first detected in Wuhan, China, in late December 2019 [1]. In March 2020, it was declared as a global pandemic and caused multidimensional life crises globally. Till the end of 2021, more than 330 million people were infected and 5.5 million deaths were recorded worldwide [2]. Additionally, it has resulted in basic health care service (HCS) utilization disparities and has featured as a major global public health concern. With the preexisting and underlying sociodemographic, clinical, and institutional factors, the pandemic has exposed individuals, society and the whole system to unwanted negative repercussions and crises [3,4,5,6,7,8]. The pandemic-related necessary HCS restructuring and lockdown restriction misery that started in March 2020 harshly reduced HCS utilization [4]. Face-to-face visits to primary and secondary health care and elective hospital admissions were forgone, postponed, or declined during the lockdown [9]. In the pandemic period, many people lost their jobs, income, access to health care, and compliance with and trust in health care [10,11,12]. These factors hindered health care institution visits and admission, even for severe medical cases during the pandemic [4,13,14,15,16,17,18,19,20]. Ray Moynihan and colleagues conducted a pooled analysis from twenty countries with over 80 studies, of which nearly 56% (55 out of 81 studies) from Europe (but none from Hungary) demonstrated that the cumulative HCS utilization, health care visits, emergency admission, and therapeutics had fallen by medians of 37.2%, 42.3%, 28.4%, and 29.6% during the pandemic compared to the prepandemic period. Additionally, they mentioned that most of the studies lack specific HCS elements, vulnerable social groups and comparisons between the two periods [21]. However, the social discrepancy in HCS use attributed to the pandemic was suggested for future research.

As mentioned above, European regions have not been immune to the impact of the COVID-19 pandemic in terms of HCS use. According to Michalowsky et al., the hospital admission rate fell by 39%, and GP and specialist visits fell by 6%, following the lockdown in Germany [4]. Similarly, the hospital admission incidence was reduced by 22% in Croatia [22], and basic and upgraded life-saving services were downgraded by 7% in Finland [8]. The GP and specialist visits postponed due to the lockdown were the highest in Portugal (55%) and reached their lowest in Bulgaria (2%), with an EU average of 26%. However, patients who missed treatments due to fear of being infected with COVID-19 during the lockdown were the highest in Israel (27%) and the lowest in Slovenia and Spain (4%), with an EU average of 12% [23]. Although it is known that most EU countries did not modify the copayment rules for medications (the proportion of costs paid by patients out of pocket) during the pandemic lockdown [9], the pandemic impact on the occurrence of patients’ inability to redeem medicine for financial reasons (cost-related prescription redemption, CRPNR) has not yet been reported [24].

There are only a few studies published so far about the health care utilization of vulnerable social groups during the pandemic. A study from South Korea showed that skipping primary HCSs during the pandemic lockdown was higher among married than single/separated/divorced subgroups, but the pandemic lockdown did not result in variations for not utilizing HCSs across the education stratum [25]. In the United Kingdom (UK), patients with all kinds of physical and mental cases showed considerable decreases in primary care contact during the first lockdown [26]. Additionally, another study from the UK revealed that ethnic minorities were more vulnerable in terms of emergency department and referral unit visits to the pandemic compared to their reference subcategories [27]. Moreover, according to the European Union Agency for Fundamental Rights, and Open Society Foundations, the health risk of the Roma population during the pandemic lockdown left them vulnerable and unable to access health care across Europe [28,29]. Furthermore, a pooled analysis of ethnic vulnerability due to the first wave pandemic lockdown revealed that the hospitalization rate was higher for ethnic minorities in the US and Europe. Hispanics, Asians, and Black African Americans were 2.08, 1.59, and 1.53 times more likely to be hospitalized than Whites, respectively [30].

In the context of Hungary, due to the increased number of new COVID-19 infections in the first wave of the epidemic, the lockdown was implemented from March to June 2020 in Hungary. Similarly, the second and third pandemic restrictions were applied from November 2020 and continued until the end of 2021 with some easing of the regulations [31]. The pandemic measures included regulated restrictions of primary, outpatient, and hospital health services to ensure the capacities for COVID-19 patient care and vaccination programs, but the lockdown restrictions were not applicable for pharmaceuticals or emergency HCSs. However, Hungary is not special in Europe with respect to the effectiveness of pandemic control, as reflected in the excess mortality data [32]. Another study revealed that sociodemographic inequity highly determined the impact of the pandemic in Hungary. The most deprived settings had a lower incidence of morbidity with higher mortality and case fatality rates [33]. Nonetheless, we did not find any investigation of the dynamics of fundamental HCS utilization attributed to the pandemic lockdown in Hungary. Furthermore, CRPNR and the pandemic impact have not yet been studied.

Our study aimed (1) to describe the prevalence of GP visits, specialist visits, hospitalizations, and CRPNR in the year before and during the COVID-19 pandemic period, (2) to determine the effect of the pandemic measures controlled for established predictors of the studied outcomes, and (3) to identify subgroups susceptible to the pandemic effect.

## 2. Materials and Methods

### 2.1. Setting

This study was a population-based, comparative cross-sectional investigation. Data for analysis were obtained from the 2021 International Social Survey Program (ISSP) [34] and the 2019 European Health Interview Survey (EHIS) Wave 3 databases of Hungary [35]. Both surveys were based on a representative sample for the whole country, and both collected data on health care use (HCU) over a one-year retrospective period.

#### 2.1.1. Data Source for Prepandemic Period

To describe the outcome parameters (i.e., the frequency of GP visits, specialist visits, hospitalization, and CRPNR) and the pertinent characteristics among adults before the pandemic, we used the third wave of the 2019 EHIS dataset of Hungary. It contained four major thematic modules on health status, health care use, lifestyle, and sociodemographic status. Data were collected by personal interviews from September 2019 to January 2020. The detailed techniques were published elsewhere [35]. The data were obtained from a representative sample of 5603 participants aged 18 years and above.

#### 2.1.2. Data Source for Pandemic Period

We used data from the Health and Health Care II panel of ISSP in 2021 to describe the HCU indicators with their determinants during the COVID-19 pandemic. The methods used for this survey were published previously [36]. This study was conducted from 15 March to 30 May 2021, during the lockdown of the third wave of COVID-19. A representative sample of 1008 Hungarian adults 18 years and above was randomly selected and interviewed in this study.

### 2.2. Outcome Variables

The GP visits in a year and specialist visits in a year were dichotomous variables for subjects who had a history of visiting their GPs and specialists, respectively, in the last 12 months prior to the survey. Similarly, hospital admission in a year was a dichotomous variable if the patient had stayed at least one night in a hospital in the last 12 months before the survey. CRPNR was a dichotomous variable defined as the respondents having missed, skipped, or replaced a prescribed drug due to financial problems at least once in the last 12 months.

### 2.3. Explanatory Variables

The primary explanatory variable in this study was the time of data collection. Participants representing prepandemic and pandemic conditions were distinguished. Outcome variables were determined for the year before the first wave of the COVID-19 pandemic by EHIS and for the 12-month period affected by the pandemic.

Education was a variable with four categories (completed grade 8 primary school, attended vocational school without a high school diploma, high school graduation with a diploma, and tertiary education included college and university graduates). The region was classified as the residential place of the subjects: Central Hungary, Central Transdanubia, Western Transdanubia, Southern Transdanubia, Northern Hungary, Northern Great Plain, and Southern Great Plain. The marital status of the participants was classified as married, single, widowed, and divorced. According to the self-reported ethnicity, the Roma and the non-Roma were distinguished (the Roma is the only large ethnic minority group in Hungary; they comprise 8.8% of the population [37]).

The age of the subjects was categorized as 18–34, 35–64, and 65 years and above. Sex was classified as male or female. The most prevalent chronic diseases, such as chronic obstructive pulmonary disease (COPD), ischemic heart disease (IHD), hypertension, diabetes mellitus, and malignancy, were registered by self-declaration of the participants as dichotomous variables.

### 2.4. Statistical Analysis

The dataset obtained by merging the two surveys was analyzed using SPSS version 21 (IBM SPSS Statistics for Windows, Version 21.0. Armonk, NY, USA: IBM Corp.). Ethical approval was not required for the secondary analysis of the anonymized data.

Bivariate analysis was performed to assess the association between the independent and outcome variables by applying logistic regression analyses. Then, multivariable logistic regression models were applied to control the confounding effect of socioeconomic status indicators (education, marital status, and ethnicity) on the outcome variables after adjusting for other independent determinants (age, region, sex, COPD, IHD, diabetes, hypertension, and malignancy) of each outcome variable. Furthermore, the interaction between the time of data collection (distinguishing prepandemic and pandemic periods) and the socioeconomic status indicators were included in the model. The GP visits, specialist visits, hospital admissions, and CRPNR within a year were the outcome variables in the four applied logistic regression models. The aim of testing for interactions was to identify the social groups vulnerable to the detrimental effect of the pandemic. The results are reported in terms of odds ratios (ORs), adjusted odds ratios (aORs), and odds ratios for interactions with time (iORs) with the corresponding 95% confidence intervals (CIs).

## 3. Results

The merged sample of this study consisted of 6611 adults. Records of participants who had no responses for corresponding outcomes were removed from the database. After cleaning for missing values for outcome variables, four distinct datasets were prepared. The sample sizes for the analysis of GP visits, specialist visits, hospital admissions, and CRPNR in a year were 6370 (N_prepandemic_ = 5368; N_pandemic_ = 1002), 6317 (N_prepandemic_ = 5323; N_pandemic_ = 994), 6408 (N_prepandemic_ = 5408; N_pandemic_ = 1000), and 5028 (N_prepandemic_ = 4337; N_pandemic_ = 691), respectively. (The sampling process is summarized in the Appendix A).

### 3.1. Sociodemographic and Clinical Characteristics

The sociodemographic composition of the prepandemic and pandemic samples showed statistically significant d ifferences. There was an overrepresentation of middle-aged individuals, women, adults with vocational or high school-level education, and Roma individuals in each pandemic sample; furthermore, Central Hungarian residents with hospital visits and CRPNR samples, and divorced individuals in each sample apart from the GP visit dataset in the pandemic sample, were detected. Concerning the clinical factors, the representation of patients with diabetes and hypertension was different between the two study periods (Table 1).

### 3.2. Outcome Measures in the Prepandemic and the COVID-19 Pandemic Periods

There were 4251 and 561 subjects who visited GPs within a year, corresponding to a prevalence of 79.2% (95% CI 78.1–80.3) and 56% (95% CI 52.9–59.1) in the prepandemic and pandemic periods, respectively. The number of participants who visited a specialist in a year was 3426 in the prepandemic period (prevalence of 64.4%, 95% CI 63.1–65.7) and 378 in the pandemic period (prevalence of 38.0%, 95% CI 35.0–41.0). A total of 728 and 68 subjects were admitted to the hospital a year before (prevalence of 13.5%, 95% CI 12.6–14.4) and during the pandemic (prevalence of 6.8%, 95% CI 5.2–8.4), respectively. These outcomes were reduced significantly during the pandemic period. As CRPNR was faced by 245 and 36 participants, the prepandemic (5.6%, 95% CI 4.9–6.3) and pandemic (5.2%, 95% CI 3.5–6.9) prevalence did not differ.

#### Factors Associated with Health Care Use and the Role of the COVID-19 Pandemic by Bivariate Analyses

Older age, female sex, and the prevalence of a chronic disease correlated with more intensive use of GPs, specialists, and hospital care but seemed to be independent of CRPNR. Patients with COPD, IHD, or diabetes faced CRPNR more frequently. Regional inequalities were observed for each outcome.

A higher level of education was associated with less intensive use of GPs but more intensive use of specialist care, as well as with less frequent hospital admissions and CRPNR admissions and CRPNR. Roma people use GPs and specialist care more frequently, and they face CRPNR more often than non-Roma people do. The occurrence of each outcome was more frequent among widowed patients and less frequent (apart from the CRPNR) among single patients. The probability of CRPNR was significantly elevated among divorced patients (Table 2). Detailed descriptive measures for each outcome in both study periods by population strata are summarized in Appendix A.

### 3.3. Determinants of Health Care Use and the Subgroup-Specific Effect of the Pandemic by Multivariable Models

The most intensive decline was observed in the probability of specialist visits and hospital admissions in the year during the pandemic. The decline in GP visit frequency was weaker and proved to be borderline significant in the multivariable model. The CRPNR showed no change during the pandemic period (Table 3).

In the multivariable logistic regression models, older age, female sex, and chronic disease proved to be factors associated with more use of HCSs. There was significant geographical variability in HCU as well.

HCU followed the general pattern among Roma: Roma ethnicity did not show a significant impact on GP visits, specialist visits, or hospital visits. However, CRPNR was more frequent (aOR = 2.018, 95% CI: 1.061–3.838) among Roma. Marital status was not dependent on the studied outcomes apart from the specialist visits in a year, which was less frequent among single (aOR = 0.753, 95% CI: 0.636–0.891) and widowed (aOR = 0.740, 95% CI: 0.597–0.918) patients. The role of higher-level education as a determinant of more frequent use of specialist care, less frequent hospital admissions, and less frequent experience with CRPNR was confirmed by a complex model. The GP visits in a year proved to be more frequent among more educated participants after controlling for the sociodemographic and clinical status of the survey participants (Table 3).

The uneven distribution of the pandemic effect by socioeconomic status was established by the interaction terms of multivariable models. The pandemic decline in hospital admission was evenly distributed by education subgroups. The pandemic effect was stronger among more educated individuals with respect to GP (iORhigh school/primary = 0.434, 95% CI: 0.243–0.776) and specialist visit (iORhigh school/primary = 0.598, 95% CI: 0.364–0.985, iORtertiary/primary = 0.331, 95% CI: 0.179–0.611) frequency. Additionally, the higher the education, the lower the CRPNR (iORhigh school/primary = 0.236, 95% CI: 0.075–0.743), as shown in the multivariable model. Considering marital status, among widowed women, general declines in GP visit (iOR = 2.284, 95% CI: 1.043–4.998) and specialist visit (iOR = 1.915, 95% CI: 1.157–3.168) frequencies were not manifested. Other significant interactions of marital status with the pandemic were not established by the applied models. As a significant interaction between Roma ethnicity and time was not confirmed in the multivariable approach, the difference between Roma and non-Roma was not demonstrated with respect to pandemic reactions in terms of HCU (Table 3).

## 4. Discussion

### 4.1. Main Findings

Through this population-based cross-sectional study, we investigated the effect of the COVID-19 pandemic on HCU in Hungary. Data analysis demonstrated a profound pandemic decrease in the GP visit frequency, specialist visit frequency, and hospital admission rate (by 23.2%, 26.4%, and 6.7%, respectively) but not in the occurrence of CRPNR when comparing the prepandemic and pandemic situations. According to publications about the use of specific health care facilities [4,22,38,39,40,41] or on general health care access [41,42], there was wide variability across European countries, and the Hungarian findings corresponded to the average of the European observations.

Hungary’s prepandemic CRPNR was within the range of published references from developed countries [43]. The observed lack of change in the pandemic period in Hungary cannot be evaluated comparatively because there are no published pandemic CRPNR results from other countries.

Our results showed that older age, female sex, and chronic disease were associated with more use of HCSs, and there were geographical inequalities in HCU. Considering social status, according to our multivariable models, the vulnerable groups are the primary educated, single, widowed, and Roma adults. These Hungarian observations are well supported in the international literature [44,45]. Altogether, the influence of sociodemographic and clinical status on HCU in Hungary has not deviated from the European mainstream.

#### 4.1.1. Pandemic Impact by Level of Education

The higher hospital admission rate of adults with primary-level education determined by our multivariable model, which was not changed during the pandemic, can be explained by their worse health status. Therefore, the observed higher frequency of GP and specialist visits among highly educated individuals cannot be explained by their poor health status and higher health care needs. Certainly, this inequality is a reflection of their differential ability and intention to use the existing services. This can be attributed to the higher proportion of nonurgent, elective actions in the medical intervention pattern of more educated individuals [21]. This gap has been significantly narrowed during the pandemic period, suggesting that elective interventions were mainly postponed. These Hungarian observations are not in line with the independence between education and HCU during the pandemic, as described in the EU [23] and the Netherlands [46], but these results were similar to the main findings of the SHARE Corona Survey on 27 European countries’ 50+ year-old populations [41]. Furthermore, our investigation demonstrated the inverse relationship of CRPNR with education, which was exaggerated in the pandemic period. Consequently, this gap has been widened.

#### 4.1.2. Pandemic Impact among Roma

The GP visit frequency was not associated with Roma ethnicity, and this relationship was not changed in the pandemic period. On the other hand, there were fewer outpatient specialist visits, and the hospital admission rate was higher among Roma patients, with a borderline significant difference. This shows that Roma individuals face serious limitations in secondary care access and that their health status is worse than that of non-Roma individuals. The CRPNR was significantly more frequent among Roma individuals, reflecting that they are overrepresented among seriously deprived individuals in Hungary. Neither of these Roma-related inequalities changed significantly during the pandemic.

A limited access to outpatient specialists and a higher hospital admission rate of Roma has been demonstrated previously in Hungary [47], and this observation is similar to numerous reports from other countries [47,48,49,50,51,52,53]. Mainly based on these former experiences and on the demonstrated increased vulnerability of racial/ethnic minorities during the COVID-19 pandemic [54], adequate monitoring among Roma individuals to avoid unequal health service delivery is recommended [29,55,56]. However, our investigation could not demonstrate the pandemic impact on ethnic inequalities in Hungary.

#### 4.1.3. Pandemic Impact by Marital Status

There was no inequality by marital status with respect to hospital admission and CRPNR in Hungary, and it was not changed during the pandemic.

In the crude analyses, the GP and specialist visit frequencies were the highest among widowed individuals, which could be attributed to the lack of informal care provided by the partner, which could have prevented the use of health services [57,58,59].

In the multivariable analyses controlled for the presence of chronic diseases, the widowed showed restricted access to specialists’ care, suggesting that their needs are not being met properly.

The general reduction in GP and specialist access during the pandemic was not manifested at all among the widowed. The interaction analysis confirmed that widows were protected against the GP and specialist access restriction; that is, the special needs of the widowed were met in the pandemic period, resulting in a narrowing of marital status inequality.

### 4.2. Practical Implications

Pandemic health care restructuring to ensure the capacities for COVID-19 patients was inevitably accompanied by serious limitations in general health care availability. According to our analysis, the impact of these restrictions on hospital use was evenly distributed across the social subgroups. However, restrictions on GPs’ and outpatient specialists’ availability did unevenly affect the social strata. Most of the observed interactions between the social status and the pandemic can be considered adequate adaptations from the viewpoint of social equity, since the prepandemic inequalities, which discriminated against the disadvantageous groups, were reduced in the pandemic period. The GPs’ and outpatient specialists’ visit reductions among highly educated adults (probably due to postponement of elective medical interventions), and the nondecreased GP and hardly decreased specialist visit frequency among the widowed patients reduced their disadvantageous status observed in the prepandemic period. The only inequality-increasing pandemic effect was observed among less educated individuals who faced more CRPNR than before the pandemic period. Altogether, health care access restraints were implemented by proper control for social consequences. The only gap widening the pandemic effect was the CRPNR increase among less educated individuals, which should be mitigated in the next phase of the epidemic.

### 4.3. Strengths and Limitations

In organizing this investigation, relevant questions on the EHIS were added to the ISSP questionnaire. This ensured the same structure of the datasets and the comparability of the two surveys’ data.

In the merged database, the prepandemic reference period was represented by a fairly large sample, which resulted in the proper statistical power for evaluating the observations from the pandemic year.

There were statistically significant but unimportant differences in the social and clinical characteristics of the samples between the two study periods. Apart from the overrepresentation of middle-aged individuals in terms of GP visits, specialist visits, and hospital admission datasets, the absolute differences in subgroup sharing were less than 10%. Although our regression models were completed with sociodemographic and clinical variables, they could not compensate for the selection bias. Taking into consideration the small absolute differences in the representation and the strengths of the significant interactions in multivariable models, selection bias could not provide an alternative explanation for the main findings about the special sensitivity of certain social groups to the pandemic impact.

In addition, the scarcity of similar topics on CRPNR during the pandemic limited us to making the further comparison with international findings. Although the out-of-pocket payment for medications issue has barely been raised in European member states [9], further importance, dynamics, and determinants during the pandemic need further investigation to establish precise inference about the target population.

The common section of the socioeconomic characteristics covered by the basic surveys (EHIS2019 and ISSP2021) determined the social subgroups we could evaluate. It is obvious that more subtle social characterization of participants would be required to establish more comprehensive conclusions.

## 5. Conclusions

We investigated the social subgroup-specific effect of the COVID-19 pandemic on HCU in Hungary by a population-based cross-sectional study implemented before and during the pandemic. We demonstrated the manifestation of established sociodemographic inequalities in our sample by the level of education, marital status, and Roma ethnicity. Our findings also show that (except for CRPNR) the COVID-19 pandemic drastically reduced the use of GP visits, outpatient specialist visits, and the hospital admission rate in Hungary, consistent with the worldwide trend.

Roma proved to be not specifically affected by the pandemic in any studied aspect. The pandemic restructuring of health care impacted the social subgroups evenly with respect to hospital care and unevenly with respect to GP and outpatient specialist visits. Primary educated and widowed patients did not follow the general trend, and their prepandemic limited HCU was not reduced further. This resulted in a pandemic-related inequality reduction. Supposing that postponing elective medical interventions—which dominated the advantageous and did not dominate the disadvantaged groups’ medical intervention patterns—explains the gap reduction; this change corresponds to the intention of pandemic regulations.

The vulnerability of primary education to CRPNR was the only gap widened in the pandemic period. This shows that although the management of pandemic HCU restrictions was implemented to avoid social inequity in Hungary, the prevention of inequity in drug availability for primary educated individuals could require more support.

## Figures and Tables

**Table 1 ijerph-19-02258-t001:** Sociodemographic and clinical characteristics of studied samples (number of cases and percentages by subgroups) in prepandemic and pandemic periods.

Explanatory Variables		GP Visits in a Year				Specialist Visits in a Year				Hospital Admission in a Year				CRPNR in a Year			
		Prepan-demic	Pan-demic	*p* ^#^	Total	Prepan-demic	Pan-demic	*p* ^#^	Total	Prepan-demic	Pan-demic	*p* ^#^	Total	Prepan-demic	Pan-demic	*p* ^#^	Total
Age groups	18–34 years	1080 (20.1)	162 (16.2)	<0.001	1242 (19.5)	1059 (19.9)	158 (15.9)	<0.001	1217 (19.3)	1094 (20.2)	161 (16.1)	<0.001	1255 (19.6)	744 (17.2)	88 (12.7)	<0.001	832 (16.5)
	35–64 years	2670 (49.7)	623 (62.2)		3293 (51.7)	2652 (49.8)	620 (62.4)		3272 (51.8)	2690 (49.7)	622 (62.2)		3312 (51.7)	2094 (48.3)	394 (57)		2488 (49.5)
	65+ years	1618 (30.1)	217 (21.7)		1835 (28.8)	1612 (30.3)	216 (21.7)		1828 (28.9)	1624 (30)	217 (21.7)		1841 (28.7)	1499 (34.6)	209 (30.2)		1708 (34)
Sex	female	2916 (54.3)	593 (59.2)	0.005	3509 (55.1)	2902 (54.5)	587 (59.1)	0.008	3489 (55.2)	2935 (54.3)	591 (59.1)	0.005	3526 (55)	2451 (56.5)	420 (60.8)	0.035	2871 (57.1)
	male	2452 (45.7)	409 (40.8)		2861 (44.9)	2421 (45.5)	407 (40.9)		2828 (44.8)	2473 (45.7)	409 (40.9)		2882 (45)	1886 (43.5)	271 (39.2)		2157 (42.9)
COPD	absent	5143 (95.8)	967 (96.5)	0.305	6110 (95.9)	5099 (95.8)	959 (96.5)	0.316	6058 (95.9)	5183 (95.8)	965 (96.5)	0.331	6148 (95.9)	4123 (95.1)	659 (95.4)	0.731	4782 (95.1)
	present	225 (4.2)	35 (3.5)		260 (4.1)	224 (4.2)	35 (3.5)		259 (4.1)	225 (4.2)	35 (3.5)		260 (4.1)	214 (4.9)	32 (4.6)		246 (4.9)
IHD	absent	5037 (93.8)	935 (93.3)	0.532	5972 (93.8)	4992 (93.8)	927 (93.3)	0.534	5919 (93.7)	5077 (93.9)	933 (93.3)	0.486	6010 (93.8)	4010 (92.5)	628 (90.9)	0.150	4638 (92.2)
	present	331 (6.2)	67 (6.7)		398 (6.2)	331 (6.2)	67 (6.7)		398 (6.3)	331 (6.1)	67 (6.7)		398 (6.2)	327 (7.5)	63 (9.1)		390 (7.8)
Hyper-tension	absent	3443 (64.1)	691 (69)	0.003	4134 (64.9)	3403 (63.9)	685 (68.9)	0.003	4088 (64.7)	3481 (64.4)	689 (68.9)	0.006	4170 (65.1)	2479 (57.2)	387 (56)	0.569	2866 (57)
	present	1925 (35.9)	311 (31)		2236 (35.1)	1920 (36.1)	309 (31.1)		2229 (35.3)	1927 (35.6)	311 (31.1)		2238 (34.9)	1858 (42.8)	304 (44)		2162 (43)
Diabetes mellitus	absent	4826 (89.9)	847 (84.5)	<0.001	5673 (89.1)	4782 (89.8)	840 (84.5)	<0.001	5622 (89)	4867 (90)	845 (84.5)	<0.001	5712 (89.1)	3811 (87.9)	539 (78)	<0.001	4350 (86.5)
	present	542 (10.1)	155 (15.5)		697 (10.9)	541 (10.2)	154 (15.5)		695 (11)	541 (10)	155 (15.5)		696 (10.9)	526 (12.1)	152 (22)		678 (13.5)
Cancer	absent	5253 (97.9)	980 (97.8)	0.915	6233 (97.8)	5210 (97.9)	972 (97.8)	0.856	6182 (97.9)	5293 (97.9)	978 (97.8)	0.883	6271 (97.9)	4228 (97.5)	672 (97.3)	0.714	4900 (97.5)
	present	115 (2.1)	22 (2.2)		137 (2.2)	113 (2.1)	22 (2.2)		135 (2.1)	115 (2.1)	22 (2.2)		137 (2.1)	109 (2.5)	19 (2.7)		128 (2.5)
Region	Central Hungary	1533 (28.6)	304 (30.3)	0.334	1837 (28.8)	1523 (28.6)	304 (30.6)	0.289	1827 (28.9)	1549 (28.6)	304 (30.4)	<0.001	1853 (28.9)	1242 (28.6)	202 (29.2)	<0.001	1444 (28.7)
	Central Transdanubia	593 (11)	110 (11)		703 (11)	590 (11.1)	110 (11.1)		700 (11.1)	595 (11)	110 (11)		705 (11)	465 (10.7)	89 (12.9)		554 (11)
	Northern Great Plain	836 (15.6)	148 (14.8)		984 (15.4)	829 (15.6)	146 (14.7)		975 (15.4)	840 (15.5)	148 (14.8)		988 (15.4)	667 (15.4)	102 (14.8)		769 (15.3)
	Northern Hungary	673 (12.5)	120 (12)		793 (12.4)	663 (12.5)	115 (11.6)		778 (12.3)	678 (12.5)	119 (11.9)		797 (12.4)	562 (13)	76 (11)		638 (12.7)
	Southern Great Plain	660 (12.3)	133 (13.3)		793 (12.4)	657 (12.3)	133 (13.4)		790 (12.5)	666 (12.3)	132 (13.2)		798 (12.5)	545 (12.6)	97 (14)		642 (12.8)
	Southern Transdanubia	508 (9.5)	86 (8.6)		594 (9.3)	502 (9.4)	86 (8.7)		588 (9.3)	513 (9.5)	86 (8.6)		599 (9.3)	413 (9.5)	53 (7.7)		466 (9.3)
	Western Transdanubia	565 (10.5)	101 (10.1)		666 (10.5)	559 (10.5)	100 (10.1)		659 (10.4)	567 (10.5)	101 (10.1)		668 (10.4)	443 (10.2)	72 (10.4)		515 (10.2)
Education	Primary	1024 (19.1)	165 (16.5)	<0.001	1189 (18.7)	1021 (19.2)	163 (16.4)	<0.001	1184 (18.7)	1026 (19)	165 (16.5)	<0.001	1191 (18.6)	871 (20.1)	140 (20.3)	0.001	1011 (20.1)
	Vocational	1291 (24)	309 (30.8)		1600 (25.1)	1282 (24.1)	308 (31)		1590 (25.2)	1296 (24)	309 (30.9)		1605 (25)	1043 (24)	215 (31.1)		1258 (25)
	High school	1817 (33.8)	398 (39.7)		2215 (34.8)	1795 (33.7)	394 (39.6)		2189 (34.7)	1839 (34)	396 (39.6)		2235 (34.9)	1448 (33.4)	252 (36.5)		1700 (33.8)
	Tertiary	1236 (23)	130 (13)		1366 (21.4)	1225 (23)	129 (13)		1354 (21.4)	1247 (23.1)	130 (13)		1377 (21.5)	975 (22.5)	84 (12.2)		1059 (21.1)
Marital status	Married	3110 (57.9)	535 (53.4)	0.077	3645 (57.2)	3084 (57.9)	534 (53.7)	<0.001	3618 (57.3)	3128 (57.8)	535 (53.5)	<0.001	3663 (57.2)	2545 (58.7)	366 (53)	<0.001	2911 (57.9)
	Single	1026 (19.1)	187 (18.7)		1213 (19)	1016 (19.1)	183 (18.4)		1199 (19)	1040 (19.2)	186 (18.6)		1226 (19.1)	709 (16.3)	100 (14.5)		809 (16.1)
	Divorced	425 (7.9)	149 (14.9)		574 (9)	423 (7.9)	147 (14.8)		570 (9)	428 (7.9)	148 (14.8)		576 (9)	354 (8.2)	104 (15.1)		458 (9.1)
	Widowed	684 (12.7)	126 (12.6)		810 (12.7)	683 (12.8)	125 (12.6)		808 (12.8)	685 (12.7)	126 (12.6)		811 (12.7)	632 (14.6)	119 (17.2)		751 (14.9)
	Missing	123 (2.3)	5 (0.5)		128 (2)	117 (2.2)	5 (0.5)		122 (1.9)	127 (2.3)	5 (0.5)		132 (2.1)	97 (2.2)	2 (0.3)		99 (2)
Ethnicity	non-Roma	5251 (97.8)	937 (93.5)	<0.001	6188 (97.1)	5209 (97.9)	930 (93.6)	<0.001	6139 (97.2)	5291 (97.8)	935 (93.5)	<0.001	6226 (97.2)	4244 (97.9)	647 (93.6)	<0.001	4891 (97.3)
	Roma	104 (1.9)	65 (6.5)		169 (2.7)	102 (1.9)	64 (6.4)		166 (2.6)	104 (1.9)	65 (6.5)		169 (2.6)	87 (2)	44 (6.4)		131 (2.6)
	Missing	13 (0.2)	(0)		13 (0.2)	12 (0.2)	(0)		12 (0.2)	13 (0.2)	(0)		13 (0.2)	6 (0.1)	(0)		6 (0.1)
Total		5368 (100)	1002 (100)	-	6370 (100)	5323 (100)	994 (100)	-	6317 (100)	5408 (100)	1000 (100)	-	6408 (100)	4337 (100)	691 (100)	-	5028 (100)

^#^ Chi-square test for prepandemic and pandemic periods between groups.

**Table 2 ijerph-19-02258-t002:** Change in the health service use in COVID-19 pandemic period compared to prepandemic period in different social strata.

	Characteristics		Prepandemic Prevalence *	Pandemic Prevalence *	OR (95%CI) **
GP visits in a year	Education level	Primary	827 (80.8)	121 (73.3)	reference
		Vocational	1026 (79.5)	176 (57.0)	**0.768 (0.641–0.920)**
		High school	1456 (80.1)	200 (50.3)	**0.753 (0.635–0.893)**
		Tertiary	942 (76.2)	64 (49.2)	**0.710 (0.590–0.855)**
	Marital status	Married	2472 (79.5)	295 (55.1)	reference
		Single	727 (70.9)	72 (38.5)	**0.612 (0.532–0.705)**
		Divorced	345 (81.2)	76 (51.0)	0.873 (0.715–1.066)
		Widowed	621 (90.8)	116 (92.1)	**3.204 (2.489–4.122)**
		Missed	86 (69.9)	2 (40.0)	NC
	Ethnicity	non-Roma	4167 (79.4)	529 (56.5)	reference
		Roma	76 (73.1)	32 (49.2)	**0.563 (0.409–0.774)**
		Missed	8 (61.5)	0 (0.0)	NC
Specialist visits in a year	Education level	Primary	614 (60.1)	84 (51.5)	reference
		Vocational	790 (61.6)	115 (37.3)	0.920 (0.790–1.071)
		High school	1177 (65.6)	145 (36.8)	1.062 (0.919–1.226)
		Tertiary	845 (69.0)	34 (26.4)	**1.288 (1.097–1.513)**
	Marital status	Married	2054 (66.6)	203 (38.0)	reference
		Single	533 (52.5)	42 (23.0)	**0.556 (0.487–0.634)**
		Divorced	293 (69.3)	49 (33.3)	0.905 (0.755–1.084)
		Widowed	475 (69.5)	82 (65.6)	**1.338 (1.136–1.576)**
		Missed	71 (60.7)	2 (40.0)	NC
	Ethnicity	non-Roma	3372 (64.7)	354 (38.1)	reference
		Roma	47 (46.1)	24 (37.5)	**0.484 (0.354–0.661)**
		Missed	7 (58.3)	0 (0.0)	NC
Hospital admission in a year	Education level	Primary	199 (19.4)	22 (13.3)	reference
		Vocational	182 (14)	24 (7.8)	**0.646 (0.526–0.795)**
		High school	207 (11.3)	17 (4.3)	**0.489 (0.400–0.598)**
		Tertiary	140 (11.2)	5 (3.8)	**0.517 (0.412–0.647)**
	Marital status	Married	404 (12.9)	31 (5.8)	reference
		Single	87 (8.4)	9 (4.8)	**0.630 (0.500–0.794)**
		Divorced	71 (16.6)	9 (6.1)	1.197 (0.926–1.547)
		Widowed	150 (21.9)	19 (15.1)	**1.953 (1.604–2.378)**
		Missed	16 (12.6)	0 (0.0)	NC
	Ethnicity	non-Roma	705 (13.3)	62 (6.6)	reference
		Roma	20 (19.2)	6 (9.2)	1.294 (0.846–1.979)
		Missed	3 (23.1)	0 (0.0)	NC
CRPNR in a year	Education level	Primary	83 (9.5)	23 (16.4)	reference
		Vocational	49 (4.7)	8 (3.7)	**0.405 (0.290–0.565)**
		High school	81 (5.6)	5 (2.0)	**0.455 (0.338–0.612)**
		Tertiary	32 (3.3)	(0.0)	**0.266 (0.177–0.399)**
	Marital status	Married	126 (5.0)	10 (2.7)	reference
		Single	44 (6.2)	4 (4.0)	1.287 (0.917–1.806)
		Divorced	25 (7.1)	9 (8.7)	**1.636 (1.108–2.415)**
		Widowed	44 (7.0)	13 (10.9)	**1.676 (1.217–2.308)**
		Missed	6 (6.2)	0 (0.0)	NC
	Ethnicity	non-Roma	230 (5.4)	28 (4.3)	reference
		Roma	14 (16.1)	8 (18.2)	**3.624 (2.254–5.828)**
		Missed	1 (16.7)	0 (0.0)	NC

* Number of cases (and proportion as %) of positive outcomes; ** odds ratios with 95% confidence intervals from logistic regression models; NC—not computable; significant results in bold.

**Table 3 ijerph-19-02258-t003:** Determinants of health care use by multivariable logistic regression models (by odds ratios with 95% confidence intervals) controlled for the interaction between pandemic and studied sociodemographic characteristics.*

Explanatory Variables		GP Visits in a Year	Specialist Visits in a Year	Hospital Admission in a Year	CRPNR in a Year
Age groups	35–64 year/18–34 year	0.954 (0.802–1.134)	0.964 (0.819–1.136)	0.969 (0.730–1.285)	0.789 (0.521–1.195)
	65+ year/18–34 year	**1.700 (1.318–2.193)**	1.112 (0.898–1.376)	**1.500 (1.089–2.068)**	0.749 (0.457–1.227)
Sex	Male/female	**0.651 (0.570–0.744)**	**0.662 (0.589–0.744)**	0.891 (0.751–1.058)	0.978 (0.744–1.285)
Chronic obstructive pulmonary disease		**2.703 (1.630–4.483)**	2.959 (2.04–4.292)	**1.979 (1.454–2.693)**	**2.104 (1.365–3.244)**
Ischemic heart disease		**1.892 (1.204–2.973)**	**3.016 (2.211–4.115)**	**2.755 (2.153–3.525)**	**1.790 (1.226–2.614)**
Hypertension		**4.039 (3.333–4.895)**	**1.830 (1.595–2.101)**	**1.366 (1.135–1.643)**	0.888 (0.658–1.198)
Diabetes mellitus		**2.841 (2.017–4.002)**	**2.546 (2.038–3.182)**	**1.566 (1.256–1.952)**	**2.104 (1.528–2.899)**
Cancer		1.746 (0.932–3.271)	**4.688 (2.591–8.483)**	**4.686 (3.24–6.776)**	1.317 (0.662–2.619)
Region	Central Transdanubia/Central Hungary	**1.547 (1.218–1.964)**	0.986 (0.809–1.202)	1.034 (0.766–1.395)	0.631 (0.384–1.036)
	Northern Great Plain/Central Hungary	1.177 (0.961–1.441)	0.865 (0.725–1.033)	1.280 (0.910–1.801)	1.301 (0.906–1.87)
	Northern Hungary/Central Hungary	1.046 (0.839–1.304)	**0.754 (0.622–0.914)**	1.060 (0.765–1.470)	1.012 (0.676–1.515)
	Southern Great Plain/Central Hungary	1.017 (0.821–1.26)	**0.735 (0.609–0.887)**	1.227 (0.878–1.714)	0.921 (0.600–1.413)
	Southern Transdanubia/Central Hungary	1.105 (0.868–1.406)	**0.752 (0.611–0.927)**	1.356 (0.970–1.894)	0.711 (0.430–1.176)
	Western Transdanubia/Central Hungary	1.250 (0.993–1.575)	0.860 (0.704–1.05)	1.178 (0.823–1.686)	**0.352 (0.184–0.673)**
Education	Vocational/Primary	**1.273 (1.008–1.606)**	**1.226 (1.014–1.481)**	0.852 (0.666–1.089)	**0.546 (0.369–0.810)**
	High School/Primary	**1.488 (1.194–1.854)**	**1.580 (1.319–1.893)**	**0.733 (0.578–0.929)**	**0.622 (0.436–0.888)**
	Tertiary/Primary	**1.258 (0.996–1.59)**	**1.797 (1.473–2.193)**	0.766 (0.587–0.999)	**0.346 (0.218–0.549)**
Marital status	Single/Married	0.990 (0.821–1.192)	**0.753 (0.636–0.891)**	0.863 (0.652–1.143)	1.217 (0.805–1.838)
	Divorced/Married	0.918 (0.697–1.210)	0.962 (0.763–1.214)	1.210 (0.905–1.618)	1.499 (0.947–2.373)
	Widowed/Married	1.113 (0.814–1.522)	**0.740 (0.597–0.918)**	1.030 (0.800–1.325)	1.091 (0.718–1.657)
Ethnicity	Roma/non-Roma	1.054 (0.647–1.718)	0.687 (0.444–1.063)	1.598 (0.924–2.763)	**2.018 (1.061–3.838)**
Year	Pandemic/Prepandemic	0.595 (0.342–1.035)	**0.459 (0.287–0.733)**	**0.480 (0.236–0.975)**	0.939 (0.371–2.376)
Education level by pandemic	Vocational/Primary	0.586 (0.325–1.057)	0.707 (0.426–1.172)	1.022 (0.485–2.154)	0.474 (0.173–1.301)
	High school/Primary	**0.434 (0.243–0.776)**	**0.598 (0.364–0.985)**	0.728 (0.328–1.614)	**0.236 (0.075–0.743)**
	Tertiary/Primary	0.536 (0.277–1.035)	**0.331 (0.179–0.611)**	0.763 (0.253–2.302)	NC
Marital status by pandemic	Single/Married	0.743 (0.497–1.11)	0.836 (0.543–1.289)	1.284 (0.556–2.962)	1.092 (0.300–3.968)
	Divorced/Married	0.831 (0.51–1.353)	0.850 (0.529–1.367)	0.822 (0.352–1.918)	2.784 (0.930–8.331)
	Widowed/Married	**2.284 (1.043–4.998)**	**1.915 (1.157–3.168)**	1.073 (0.52–2.215)	2.009 (0.716–5.642)
Ethnicity by pandemic	Roma/non-Roma	0.480 (0.207–1.112)	1.130 (0.511–2.497)	0.533 (0.168–1.687)	0.827 (0.251–2.725)

* Significant results in bold.

## Data Availability

The database of EHIS 2019 is available from the Hungarian Central Statistical Office through a registration process (https://www.ksh.hu/data_access_safe_centre_access, accessed on 24 December 2021). Data downloadable version of the International Social Survey Program 2021 is under preparation (http://www.issp.org/data-download/by-year/, accessed on 24 December 2021). The database processed in the investigation is available from the corresponding author upon request.

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
