# Peer review of "The Effect of the COVID-19 Pandemic on the Social Inequalities of Health Care Use in Hungary: A Nationally Representative Cross-Sectional Study"

_ijerph, 2022, doi:10.3390/ijerph19042258_

Round 1
Reviewer 1 Report
The article deals with an interesting topic, the general idea is fascinating and conducted across a setting where evidences are not that much, so able potentially to add value around the (well-known) covid19 health service performance topic. However,
The abstract is too long, not easy to read, and does not properly cover the proposed work's main aspects. The abstract' goal is to serve, in a concise way, the readers to get an overall idea of the investigated topic and determine the relevance of the general findings communicating those results effectively. The current version fails in both aspects. The abstract must be reduced and reorganized.
The introduction is complicated to read and not well-connected with the rest of the manuscript making it complicated to understand what the authors want to investigate. More important, it seems that some concepts are “taken for granted” in the description which, instead, require more explanation and references. In details:
- Page 2; line 79: Authors reported: Authors reported: “the pandemic impact on medication has not yet been reported.”; Medication what? Adherence? Cost? Access? Supplies? Please clarify this section.
- Pages 2-3; lines 98-99: Authors reported: “November 2020 and continued until now with some”; Please provide a date! “until now” cannot be considered.
- Can the authors explain what they meant by: “tertiary health services”? (page 3; line 100).
- The concept of: “cost-related prescription non-redemption (CRPNR)” falls from the sky, without any previous (and after), proper explanation, introduction and references’ support. Despite the centrally of the concept among the study aim, such concept isn’t detailed and considered in the overall implication and study hypothesis development.
Below are other concerns that must be addressed.
- In which database the CRPNR data are stored?
- Page 4, lines 188-190; Isn’t’ clear what the authors meant. Is this the sample size selection? description?
- The result section is a bit overlooked. It is pretty messy organized and complicated to detect how the selected variables interact with each other and the overall effects size related to the model fit execution.
- The results section is well defined, but I still cannot understand the study's goals. Is this a kind of subgroup analysis of minority group health inequality? Why not consider other minority groups? Or, on the other hand, perhaps the Hungarian CRPNP trend, compared across the socio-demographic group the main subject under investigation?
In my view, the article requires substantial reorganization and a precise formulation of the research question the authors want to prove.
Author Response
Reviewer 1
Dear Reviewer,
Thank you very much for the careful review of our manuscript. Each comment and suggestion has been considered. The corresponding changes and refinements made in the revised paper are summarized in our response.
Regarding the English language, the manuscript was edited by American Journal Experts for grammar and related language issues before the submission. (The certification was uploaded along with the manuscript.)
All the changes in the manuscript are shown with yellow highlighting in the manuscript.
Answers along with the modifications we made are summarized below (comments/questions of Yours are in capitals).
Sincerely yours, Janos Sandor (on behalf of the authors)
1.
THE ARTICLE DEALS WITH AN INTERESTING TOPIC, THE GENERAL IDEA IS FASCINATING AND CONDUCTED ACROSS A SETTING WHERE EVIDENCE ARE NOT THAT MUCH, SO ABLE POTENTIALLY TO ADD VALUE AROUND THE (WELL-KNOWN) COVID-19 HEALTH SERVICE PERFORMANCE TOPIC.
THE ABSTRACT IS TOO LONG, NOT EASY TO READ, AND DOES NOT PROPERLY COVER THE PROPOSED WORK'S MAIN ASPECTS. THE ABSTRACT' GOAL IS TO SERVE, IN A CONCISE WAY, THE READERS TO GET AN OVERALL IDEA OF THE INVESTIGATED TOPIC AND DETERMINE THE RELEVANCE OF THE GENERAL FINDINGS COMMUNICATING THOSE RESULTS EFFECTIVELY. THE CURRENT VERSION FAILS IN BOTH ASPECTS. THE ABSTRACT MUST BE REDUCED AND REORGANIZED.
The abstract has shortened and corrected according to the suggestions. The length of original and corrected abstract was 2447 and 2000 characters, respectively. (The modifications are signed in the revised manuscript.)
2.
THE INTRODUCTION IS COMPLICATED TO READ AND NOT WELL-CONNECTED WITH THE REST OF THE MANUSCRIPT MAKING IT COMPLICATED TO UNDERSTAND WHAT THE AUTHORS WANT TO INVESTIGATE. MORE IMPORTANT, IT SEEMS THAT SOME CONCEPTS ARE “TAKEN FOR GRANTED” IN THE DESCRIPTION WHICH, INSTEAD, REQUIRE MORE EXPLANATION AND REFERENCES. IN DETAILS:
We have made crucial modifications accordingly. These modifications were well articulated, important references were cited and could express the idea of the problem under the study.
Paragraphs were merged and reduced accordingly. The concepts raised have been addressed up to the required points.
In the present form, we started with the whole notion of the key element COVID-19 pandemic→ recent epidemiological and health care service (HCS) utilization impact across the globe →showed an overview of affected social and system sections in the first paragraph. Previously two paragraphs [1st and 2nd] are now merged into one paragraph in a well-focused manner.
In the second paragraph, we have aimed to show the previous magnitude and impact of the pandemic on the studied outcomes in European and other contexts from relevant literature. Efforts were made to show the missing gaps. After careful consideration and moderate edition, we brought 3rd paragraph into the second position.
The third paragraph dealt with the vulnerable social groups to HCS use in terms of marital status, education and ethnicity. Because our impression or hypothesis was whether the pandemic has a sole effect on these strata and if so, the target interventions will be taken. From this point of view, we have come across the previous literature and tried to show them briefly in terms of pandemic effects on the studied outcome variables and the strata's role. The most important point here is that these sociodemographic factors have unlimited importance on HCS utilization before the pandemic across Europe. https://doi.org/10.1016/j.ehb.2021.101049 https://doi.org/10.1186/1471-2458-10-224 https://www.mdpi.com/1660-4601/17/7/2602 https://www.ncbi.nlm.nih.gov/books/NBK459030/ However, during the pandemic, the dynamics of their relevance on HCS use has not been well investigated. That initiated us to demonstrate the interaction of the pandemic on HCS utilization in terms of GP visits, specialist care, hospital admission rate and CRPNR among these strata in Hungary. Additionally, we have made a few modifications here.
The fourth paragraph is all about the present phenomenon in Hungary context per the problem under the study. Lastly, the final paragraph dealt with the objectives.
Therefore, we hope that we found your suggestions and you are satisfied with our reasonable reactions.
3.
PAGE 2; LINE 79: AUTHORS REPORTED: AUTHORS REPORTED: “THE PANDEMIC IMPACT ON MEDICATION HAS NOT YET BEEN REPORTED.”; MEDICATION WHAT? ADHERENCE? COST? ACCESS? SUPPLIES? PLEASE CLARIFY THIS SECTION.
The correction has been made. [Line 76].
original version:
”the pandemic impact on medication has not yet been reported.”
corrected version:
”the pandemic impact on cost-related prescription nonredemption (CRPNR) has not yet been reported.”
4.
PAGES 2-3; LINES 98-99: AUTHORS REPORTED: “NOVEMBER 2020 AND CONTINUED UNTIL NOW WITH SOME”; PLEASE PROVIDE A DATE! “UNTIL NOW” CANNOT BE CONSIDERED.
The correction has been made. [Line 96].
original version:
”continued until now…”
corrected version:
”continued until the end of 2021…”
5.
CAN THE AUTHORS EXPLAIN WHAT THEY MEANT BY: “TERTIARY HEALTH SERVICES”? (PAGE 3; LINE 100).
Tertiary health care meant the curative services provided by a specialist at higher health institutions mostly hospitals. This terminology is rather specific for Hungarian setting. To avoid the term misleading for international readers, the text had been modified. [Line 97-98].
original version:
”restrictions of primary, secondary, and tertiary health services…”
corrected version:
”restrictions of primary, outpatient, and hospital health services…”
6.
THE CONCEPT OF: “COST-RELATED PRESCRIPTION NON-REDEMPTION (CRPNR)” FALLS FROM THE SKY, WITHOUT ANY PREVIOUS (AND AFTER), PROPER EXPLANATION, INTRODUCTION AND REFERENCES’ SUPPORT. DESPITE THE CENTRALLY OF THE CONCEPT AMONG THE STUDY AIM, SUCH CONCEPT ISN’T DETAILED AND CONSIDERED IN THE OVERALL IMPLICATION AND STUDY HYPOTHESIS DEVELOPMENT.
From the very beginning, it has been our big concern. For this reason, we tried to quest for the relevant literature on this variable during the pandemic period. But we found only the bare information about the out-of-pocket payment which has a similar notion/concept with CRPNR in the second paragraph. That was …. Baudrant-Boga and colleagues …stated that “that most EU countries did not modify out-of-pocket payments for medication during the pandemic lockdown”. [Line 75]. But no further details about the countries in the EU and the extent of CRPNR were stated. So, we believed that our hypothesis and final finding will be a baseline for future investigation. Furthermore, there are few studies on general prescription non-redemption in Hungary but the effect of medications cost has not yet been investigated.
7.
BELOW ARE OTHER CONCERNS THAT MUST BE ADDRESSED.
IN WHICH DATABASE THE CRPNR DATA ARE STORED?
The database of the European Health Interview Survey 2019 (EHIS) is stored by the Hungarian Central Statistical Office (HCSO) and is available through a registration process (https://www.ksh.hu/data_access_safe_centre_access). The database of the International Social Survey Program 2021 (ISSP) is stored in the ISSP website (http://www.issp.org/data-download/by-year/). The dataset is under preparation now. However, the database processed in our investigation is available for readers upon reasonable request from the corresponding author.
To make more explicit the data availability, the Data Availability Statement has been modified. [Line 413-417]
original version:
”All information regarding the current study is available on the ISSP website unless the embargo period limits. While EHIS data is available from the Hungarian Central Statistical Office based on reasonable request.”
corrected version:
”The database of EHIS 2019 is available from the Hungarian Central Statistical Office through a registration process (https://www.ksh.hu/data_access_safe_centre_access). Data downloadable version of the International Social Survey Program 2021 is under preparation (http://www.issp.org/data-download/by-year/). The database processed in the investigation is available from the corresponding author upon request.”
8.
PAGE 4, LINES 188-190; ISN’T’ CLEAR WHAT THE AUTHORS MEANT. IS THIS THE SAMPLE SIZE SELECTION? DESCRIPTION?
The sampling process is summarized by the figure referred by the sentence. The obviously wrong sentence has been reformulated. [Line 183].
original version:
”See the appendix of sample size selection.”
corrected version:
”(The sampling process is summarized in the Appendix Figure S1.)”
9.
THE RESULT SECTION IS A BIT OVERLOOKED. IT IS PRETTY MESSY ORGANIZED AND COMPLICATED TO DETECT HOW THE SELECTED VARIABLES INTERACT WITH EACH OTHER AND THE OVERALL EFFECTS SIZE RELATED TO THE MODEL FIT EXECUTION.
THE RESULTS SECTION IS WELL DEFINED, BUT I STILL CANNOT UNDERSTAND THE STUDY'S GOALS. IS THIS A KIND OF SUBGROUP ANALYSIS OF MINORITY GROUP HEALTH INEQUALITY? WHY NOT CONSIDER OTHER MINORITY GROUPS? OR, ON THE OTHER HAND, PERHAPS THE HUNGARIAN CRPNP TREND, COMPARED ACROSS THE SOCIO-DEMOGRAPHIC GROUP THE MAIN SUBJECT UNDER INVESTIGATION?
IN MY VIEW, THE ARTICLE REQUIRES SUBSTANTIAL REORGANIZATION AND A PRECISE FORMULATION OF THE RESEARCH QUESTION THE AUTHORS WANT TO PROVE.
a.
In order to improve the readability of the Results section, two subtitles were modified and an explanatory sentence has been extended.
original version (Line 207-208):
“3.2.1 Factors associated with health care use and the role of the COVID-19 pandemic”
corrected version:
“3.2.1 Factors associated with health care use and the role of the COVID-19 pandemic by bivariate analyses”
original version (Line 226-227):
“3.3 Determinants of health care use and the subgroup-specific effect of the pandemic”
corrected version:
“3.3 Determinants of health care use and the subgroup-specific effect of the pandemic by multivariable models”
original version (Line 123):
“The uneven distribution of the pandemic effect by socioeconomic status was established by multivariable models.”
corrected version:
“The uneven distribution of the pandemic effect by socioeconomic status was established by the interaction terms of multivariable models.”
b.
The common section of the socioeconomic characteristics covered by the basic surveys (EHIS2019 and ISSP2021) determined the social subgroups we could evaluate. It is obvious that analyzing more social subgroups could improve the value of our investigation. It has been acknowledged by inserting a new sentence into the Strengths and limitations section. (Line 366-369)
inserted sentence:
“The common section of the socioeconomic characteristics covered by the basic surveys (EHIS2019 and ISSP2021) determined the social subgroups we could evaluate. It is obvious that more subtle social characterization of participants would be required to establish more comprehensive conclusions.”
c.
The last paragraph of the Introduction section with declaration of objectives has been modified to define more precisely the aims. (Line 107-110)
original version:
“Therefore, we found that it is indispensable to investigate the effect of the COVID-19 pandemic together with other explanatory characteristics on GP visits, specialist visits, hospitalizations and CRPNR in Hungary, which has not been evaluated in previous studies. In detail, we (1) scrutinized the prevalence of GP visits, specialist visits, hospitalizations and CRPNR before and during the lockdown of the COVID-19 pandemic, (2) investigated the effect of the pandemic on CRPNR GP visits, specialist visits and hospitalizations controlled for established predictors, and (3) determined subgroups susceptible to GP visits, specialist visits, hospitalizations and CRPNR elicited by the pandemic.”
corrected version:
“Our study aimed (1) to describe the prevalence of GP visits, specialist visits, hospitalizations and CRPNR in a year before and during the COVID-19 pandemic period, (2) to determine the effect of the pandemic measures controlled for established predictors of the studied outcomes, and (3) to identify subgroups susceptible to pandemic effect.”
Reviewer 2 Report
Overall: The major contribution of this paper is to use four multivariable logistic regression models to analyze the Hungary national data for prepandemic and pandemic GP visit, specialist visit, hospital admission and cost related prescription nonredemption to find out whether there are social inequality gaps from pre-pandemic to during pandemic from a quantitative perspective regarding age, education levels, marital status, region, and ethnicity group.
Comments:
Line 33-34: What is the difference between two results? Because the names are the same: two “ORhigh-school vs primary-education” and two “ORwidowed vs married”? Maybe should mention “before and during pandemic” in the text to distinguish the same names because I firgure out only after reading through the tables in the paper to figure out the different numbers for different names. Also, should that be pandemic effect was "stronger" not "weaker" for "high-school/primary-education" group? The OR for this group dropped significantly for this group pre- and during- pandemic for GP visit (prepandemic OR: 1.488, during pandemic OR: 0.434) and specialist visit(prepandemic OR: 1.58, during pandemic OR: 0.598) in Table 3. My understanding is that because of the pandemic, the "high-school/primary education" group becomes less accessible to GP and specialist visits, therefore we see the drop in OR from pre-pandemic to during-pandemic.
Line 220-221: From numbers reported in Table2, higher level of education seems to be associated with more intensive use of specialist care and less associated with GP visit, not "higher level of education was associated with more intensive use of GP and specialist visit." Please verify.
Line 227: Can we also look at the appendix for data transformation techniques?
Line 241: Should that be "Roma did not show a significant impact of specialist visits" because we see a rise of OR(from below 1 to above 1, that means Roma is accessible to specialist visit during-pandemic more than they could pre-pandemic, therefore the inequality for Roma to visit specialist is not widened) according to the OR[prepandemic OR:0.687, during-pandemic OR:1.13 ] from Table 2? GP and Hospital visits seem to be impacted (dropped) from pre-pandemic compared to during-pandemic for Roma ORs. However, the 95%CI for GP, specialist, and hospital admissions seem inconclusive covering both above 1 and below 1 area?
Table 3: 18-34 years are in two age groups or is it a ratio? A ratio of male/female? Are all variables ratios? Especially on the “education level by pandemic”, “marital status by pandemic” and “ethnicity by pandemic”? Also, can we look at the corresponding Appendix for data transformation methods? Overall, it did not report the P-values in the four multiple logistic regression models. According to US national library of medicine national institutes of health, it is important to note that, unlike p-value, 95% CI does not report a measure’s statistical significance. https://www.ncbi.nlm.nih.gov/pmc/articles/PMC2938757/.
Author Response
Reviewer 2
Dear Reviewer,
Thank you very much for the careful review of our manuscript. Each comment and suggestion has been considered. The corresponding changes and refinements made in the revised paper are summarized in our response.
Regarding the English language, the manuscript was edited by American Journal Experts for grammar and related language issues before the submission. (The certification was uploaded along with the manuscript.)
All the changes in the manuscript are shown with yellow highlighting in the manuscript.
Answers along with the modifications we made are summarized below (comments/questions of Yours are in capitals).
Sincerely yours, Janos Sandor (on behalf of the authors)
1.
OVERALL: THE MAJOR CONTRIBUTION OF THIS PAPER IS TO USE FOUR MULTIVARIABLE LOGISTIC REGRESSION MODELS TO ANALYZE THE HUNGARY NATIONAL DATA FOR PREPANDEMIC AND PANDEMIC GP VISIT, SPECIALIST VISIT, HOSPITAL ADMISSION AND COST RELATED PRESCRIPTION NONREDEMPTION TO FIND OUT WHETHER THERE ARE SOCIAL INEQUALITY GAPS FROM PRE-PANDEMIC TO DURING PANDEMIC FROM A QUANTITATIVE PERSPECTIVE REGARDING AGE, EDUCATION LEVELS, MARITAL STATUS, REGION, AND ETHNICITY GROUP.
LINE 33-34: WHAT IS THE DIFFERENCE BETWEEN TWO RESULTS? BECAUSE THE NAMES ARE THE SAME: TWO “ORHIGH-SCHOOL VS PRIMARY-EDUCATION” AND TWO “ORWIDOWED VS MARRIED”? MAYBE SHOULD MENTION “BEFORE AND DURING PANDEMIC” IN THE TEXT TO DISTINGUISH THE SAME NAMES BECAUSE I FIRGURE OUT ONLY AFTER READING THROUGH THE TABLES IN THE PAPER TO FIGURE OUT THE DIFFERENT NUMBERS FOR DIFFERENT NAMES. ALSO, SHOULD THAT BE PANDEMIC EFFECT WAS "STRONGER" NOT "WEAKER" FOR "HIGH-SCHOOL/PRIMARY-EDUCATION" GROUP? THE OR FOR THIS GROUP DROPPED SIGNIFICANTLY FOR THIS GROUP PRE- AND DURING- PANDEMIC FOR GP VISIT (PREPANDEMIC OR: 1.488, DURING PANDEMIC OR: 0.434) AND SPECIALIST VISIT(PREPANDEMIC OR: 1.58, DURING PANDEMIC OR: 0.598) IN TABLE 3. MY UNDERSTANDING IS THAT BECAUSE OF THE PANDEMIC, THE "HIGH-SCHOOL/PRIMARY EDUCATION" GROUP BECOMES LESS ACCESSIBLE TO GP AND SPECIALIST VISITS, THEREFORE WE SEE THE DROP IN OR FROM PRE-PANDEMIC TO DURING-PANDEMIC.
The confusing structure of the criticized sentence was corrected along with an insertion into the Methods section to emphasize that the main result of our study are the ORs for interaction terms in the multivariable models. Thanks for identifying this structural problem and suggesting the modification.
original version (Line 24-25):
“To identify the social strata susceptible to pandemic effect, the interaction of the time of data collection with level of education, marital status, and ethnicity, was also tested.”
corrected version:
“To identify the social strata susceptible to pandemic effect, the interaction of the time of data collection with level of education, marital status, and ethnicity, was also tested and described by iORs.”
original version (Line 29-33):
“However, the pandemic effect was weaker among primary educated adults (ORhigh-school vs primary-education =0.434; 95% CI 0.243-0.776, ORhigh-school vs primary-education =0.598; 95% CI 0.364-0.985), and among widows (ORwidowed vs married =2.284; 95% CI 1.043-4.998, ORwidowed vs married=1.915; 95% CI 1.157-3.168) on the frequency of GP visit and specialist visit; and the prepandemic CRPNR inequality by level of education was increased (ORhigh-school vs primary-education =0.236; 95% CI 0.075-0.743).”
corrected version:
“However, the pandemic effect was weaker among primary educated adults (iORGP visit; high-school vs primary-education =0.434; 95% CI 0.243-0.776, iORspeciaist visit; high-school vs primary-education =0.598; 95% CI 0.364-0.985), and stronger among married (iORGP visit; widowed vs married =2.284; 95% CI 1.043-4.998, iORspecialist visit; widowed vs married=1.915; 95% CI 1.157-3.168) on the frequency of GP visit and specialist visit; and the prepandemic CRPNR inequality by level of education was increased (iORhigh-school vs primary-education =0.236; 95% CI 0.075-0.743).”
2.
LINE 220-221: FROM NUMBERS REPORTED IN TABLE2, HIGHER LEVEL OF EDUCATION SEEMS TO BE ASSOCIATED WITH MORE INTENSIVE USE OF SPECIALIST CARE AND LESS ASSOCIATED WITH GP VISIT, NOT "HIGHER LEVEL OF EDUCATION WAS ASSOCIATED WITH MORE INTENSIVE USE OF GP AND SPECIALIST VISIT." PLEASE VERIFY.
We have to admit that result presented in the Table 2 is wrongly mentioned in the text. Thank you very much for identifying our mistake. A modification has been made accordingly. [Line 213-215]
original version:
”A higher level of education was associated with more intensive use of GPs and specialist care, as well as with less frequent hospital admissions and CRPNR.”
corrected version:
”A higher level of education was associated with less intensive use of GPs but more intensive use of specialist care, as well as with less frequent hospital admissions and CRPNR.”
3.
LINE 227: CAN WE ALSO LOOK AT THE APPENDIX FOR DATA TRANSFORMATION TECHNIQUES?
We have got the required data from the authorized bodies. Then each variable was recoded (using common categories could be classified by both EHIS2019 and ISSP2021 data collection systems) as a new variable in the SPSS including the type and measurement of the variable. After recoding each variable the original names were removed from the SPSS. In this way, the data were transformed. Since we have two separate databases with a different number of cases and extra variables, we have selected common variables that exist in both datasets [EHIS 2019-pre-pandemic and ISSP 2021-pandemic databases]. After finalizing the selection of those common and important variables, we have created a new database by merging two datasets by variables. Through this process, the completeness of each variable has been confirmed and checked before analysis. All of our outcome variables are dummy variables and the interpretation of the results is made accordingly.
Because the surveys’ methodology is published and cited in our paper, and the process of data conversation applied only basic methods, the text of the manuscript was not modified. We hope that you can accept this approach.
4.
LINE 241: SHOULD THAT BE "ROMA DID NOT SHOW A SIGNIFICANT IMPACT OF SPECIALIST VISITS" BECAUSE WE SEE A RISE OF OR(FROM BELOW 1 TO ABOVE 1, THAT MEANS ROMA IS ACCESSIBLE TO SPECIALIST VISIT DURING-PANDEMIC MORE THAN THEY COULD PRE-PANDEMIC, THEREFORE THE INEQUALITY FOR ROMA TO VISIT SPECIALIST IS NOT WIDENED) ACCORDING TO THE OR [PREPANDEMIC OR: 0.687, DURING-PANDEMIC OR:1.13] FROM TABLE 2? GP AND HOSPITAL VISITS SEEM TO BE IMPACTED (DROPPED) FROM PRE-PANDEMIC COMPARED TO DURING-PANDEMIC FOR ROMA ORS. HOWEVER, THE 95%CI FOR GP, SPECIALIST, AND HOSPITAL ADMISSIONS SEEM INCONCLUSIVE COVERING BOTH ABOVE 1 AND BELOW 1 AREA?
The results mentioned in the 3.3 subsection (Determinants of health care use and the subgroup-specific effect of the pandemic) are from the multivariable logistic regression models shown in Table 3.
The OR=0.687 [95%CI 0.444-1.063] shows that in the whole dataset covering both prepandemic and pandemic period the Roma ethnicity shoes no significant association with the frequency of outpatient visits.
The OR=1.13 [95%CI 0.511-2.497] shows that there is no significant interaction between Roma ethnicity and time of data collection (i.e. there is no evidence from the analysis that the prepandemic use of outpatient care among Roma changed during the pandemic period).
Table 2 present the descriptive statistics for prepandemic and pandemic periods and the association between explanatory variables and outcomes by bivariate analysis.
To improve the clarity of the presentation of results we modified the subtitles:
original version (Line 207-208):
“3.2.1 Factors associated with health care use and the role of the COVID-19 pandemic”
corrected version:
“3.2.1 Factors associated with health care use and the role of the COVID-19 pandemic by bivariate analyses”
original version (Line 226-227):
“3.3 Determinants of health care use and the subgroup-specific effect of the pandemic”
corrected version:
“3.3 Determinants of health care use and the subgroup-specific effect of the pandemic by multivariable models”
Further, the explanatory sentences in this section was also modified (Line 246):
original version:
“The uneven distribution of the pandemic effect by socioeconomic status was established by multivariable models.”
corrected version:
“The uneven distribution of the pandemic effect by socioeconomic status was established by the interaction terms of multivariable models.”
5.
TABLE 3: 18-34 YEARS ARE IN TWO AGE GROUPS OR IS IT A RATIO? A RATIO OF MALE/FEMALE? ARE ALL VARIABLES RATIOS? ESPECIALLY ON THE “EDUCATION LEVEL BY PANDEMIC”, “MARITAL STATUS BY PANDEMIC” AND “ETHNICITY BY PANDEMIC”? ALSO, CAN WE LOOK AT THE CORRESPONDING APPENDIX FOR DATA TRANSFORMATION METHODS?
The explanatory variables used in our analyses are described in the 2.3 Explanatory variables section.
Regarding Table 3, it means that age group variable has three categories [18-34, 35-64 and 65+ years] and 18-34 years is the reference category in regression models. The “35-64 year / 18-34 year” is not a ratio but a notion that the participants 35-64 years old are compared to the reference of the participants 18-34 years old. Similarly for other explanatory variables. The reason why we followed such a way of writing was to minimize the size of the table.
The Table 3 was not restructured (the reference categories were not displayed in distinct rows with mentioning that those categories are the references and the OR is 1 for those). We hope that You can accept our approach.
6.
OVERALL, IT DID NOT REPORT THE P-VALUES IN THE FOUR MULTIPLE LOGISTIC REGRESSION MODELS. ACCORDING TO US NATIONAL LIBRARY OF MEDICINE NATIONAL INSTITUTES OF HEALTH, IT IS IMPORTANT TO NOTE THAT, UNLIKE P-VALUE, 95% CI DOES NOT REPORT A MEASURE’S STATISTICAL SIGNIFICANCE. HTTPS://WWW.NCBI.NLM.NIH.GOV/PMC/ARTICLES/PMC2938757/.
We used the 95% confidence intervals as it is written in the cited paper
“What about confidence intervals?
The 95% confidence interval (CI) is used to estimate the precision of the OR. A large CI indicates a low level of precision of the OR, whereas a small CI indicates a higher precision of the OR. It is important to note however, that unlike the p value, the 95% CI does not report a measure’s statistical significance. In practice, the 95% CI is often used as a proxy for the presence of statistical significance if it does not overlap the null value (e.g. OR=1). Nevertheless, it would be inappropriate to interpret an OR with 95% CI that spans the null value as indicating evidence for lack of association between the exposure and outcome.”
If the 95%CI does not include 1 than we considered that the point estimates’ deviation from 1 cannot be explained by chance applying the threshold of 0.05 probability.
If the 95%CI does include 1 than we considered the OR as a measure of association which does not demonstrate in our analysis that the explanatory factor influences the outcome. We aware that this result cannot distinguish between the association and the type 2 error.
Checking the text we could find the following sentences with not fully correct interpretation. We modified those accordingly.
original version (Line 237-238):
“Marital status was independent of the studied outcomes apart from the specialist visits in a year …)
corrected version:
“Marital status was not dependent of the studied outcomes apart from the specialist visits in a year …)
original version (Line 257-258):
“the Roma and non-Roma were no different with respect to pandemic reactions in terms of health care use …”
corrected version:
“the difference between Roma and non-Roma was not demonstrated with respect to pandemic reactions in terms of health care use …”
Reviewer 3 Report
Dear Authors
Greetings
Please follow the suggestions on the attached doc, reading it with Adobe.
Kind regards

Author Response
Reviewer 3
Dear Reviewer,
Thank you very much for the careful review of our manuscript. Each comment and suggestion has been considered. The corresponding changes and refinements made in the revised paper are summarized in our response.
Regarding the English language, the manuscript was edited by American Journal Experts for grammar and related language issues before the submission. (The certification was uploaded along with the manuscript.)
All the changes in the manuscript are shown with yellow highlighting in the manuscript.
Answers along with the modifications we made are summarized below (comments/questions of Yours are in capitals).
Sincerely yours, Janos Sandor (on behalf of the authors)
1.
THE TITLE IS VERY LONG. I SUGGEST: A CROSS-SECTIONAL OF COVID-19'S IMPACTS ON COST-RELATED PRESCRIPTION NON-REDEMPTION, HEALTH CARE SEEKING AND HOSPITALIZATION IN HUNGARY
We have to admit that the original title needed improvement. A shorter version has been formulated with explicit mention of the main focus of the study, the pandemic influence on social inequalities.
original version (177 characters):
”Effect of the COVID-19 Pandemic on Cost-Related Prescription Nonredemption, Health care Seeking and Hospitalization in Hungary: A Nationally Representative Cross-Sectional Study”
corrected version (129 characters):
”The COVID-19 Pandemic on the Social Inequalities of Health Care Use in Hungary: A Nationally Representative Cross-Sectional Study”
2.
THE AUTHORS CAN WRITE A SHORTER INTRODUCTION BEING MORE OBJECTIVE ACCORDING TO THE AIMS OF THE ARTICLE.
We have made extensive modifications, particularly for the first two paragraphs as they need too much concentration on the topic of the study.
3.
THE AUTHORS WOULD CAN PROVIDE DATA SOURCE OF ONE YEAR BEFORE (MARCH TO MAY 2019) AND (MARCH TO MAY 2020) FOR COMPLETE COMPARATIVE ANALYSIS.
The two surveys and their datasets are described in two distinct subsections (2.1.1 and 2.1.2) Unfortunately, the titles of the subsections were not properly formulated. Both have been corrected. [Line 118, 126]
original version:
”2.1.1 Before the COVID-19 pandemic data source … 2.1.2 The data source during the COVID-19 pandemic”
corrected version:
”2.1.1 Data source for prepandemic period … 2.1.2 Data source for pandemic period”
4.
THE AUTHORS WOULD CAN PROVIDE GRAPHICAL ANALYSIS, ENABLES CLARITY OF COMMUNICATION AND PROVIDES FOCUS FOR FURTHER ANALYSIS. IT IS AN IMPORTANT TOOL FOR UNDERSTANDING SOURCES OF VARIATION IN THE DATA AND THEREBY HELPING TO BETTER UNDERSTAND THE PROCESS AND WHERE ROOT CAUSES MIGHT BE.
Now we have attached a figure to the appendix of how we have reached the final sample step by step for each outcome variable. The reference for the figure has been inserted int the text. [Line 183 and the Appendix Figure S1]
inserted sentence:
“(The sampling process is summarized in the Appendix Figure S1.)”
5.
ROMA IN THE COVID-19 CRISIS – RELIEFWEB
https://reliefweb.int/sites/reliefweb.int/files/resources/Roma%20in%20the%20COVID-19%20crisis%20-%20An%20early%20warning%20from%20six%20EU%20Member%20States.pdf
Thank you very much for the suggestion of this publication! This has been cited under the introduction section. [Line 89 and under the reference section Line 500, the reference number 28]
6.
AND THE OUTCOMES????????
If we found your point here, this question is related to the Conclusions section of the manuscript.
In this section, we have tried to make a precise conclusion by displaying the most important findings that were demonstrated in our study. It is just to show brief inferences for the readers. But others have been included in the whole text under the result section.
Conclusion section was not modified. We hope that it can be accepted by You.
7.
https://www.google.com/search?q=gypsies+covid+19&rlz=1C1CHBF_en&sxsrf=AOaemvJHcMFUNp1_u-0eo9irKzXuRrJQ6g%3A1642374397839&ei=_aTkYfXYMq-Jxc8PsY-EaA&ved=0ahUKEwi1y8i0sbf1AhWvRPEDHbEHAQ0Q4dUDCA4&uact=5&oq=gypsies+covid+19&gs_lcp=Cgdnd3Mtd2l6EAM6BAgjECc6EQguEIAEELEDEIMBEMcBENEDOggIABCABBCxAzoICAAQsQMQgwE6BQgAEIAEOgcILhCxAxBDOg4ILhCABBCxAxDHARDRAzoICC4QgAQQsQM6EQguEIAEELEDEIMBEMcBEK8BOgQIABBDOgsILhCABBDHARCvAToFCC4QgAQ6BwguEIAEEAo6BAgAEAo6BQgAEMsBOgcIABCABBAKOgQILhAKOggIABAWEB4QE0oECEEYAEoECEYYAFAAWKEsYJwxaABwAXgAgAF1iAHvC5IBBDExLjWYAQCgAQHAAQE&sclient=gws-wiz
Regarding this Google search link for “gypsies covid 19” terms, we believe that citing the above literature work adds its input. But we used the most relevant literature mainly journal articles regarding COVID-19 pandemic versus healthcare utilization disparities. Because the majority of the literature were opinions, comments, newspapers, magazine sources, shallow ideas about the problem under study. These pushed us to focus on internationally peer-reviewed journal articles. But this does not mean we did not use other sources of literature as evidence in our study.
References were not extended by results of the suggested search. We hope that it can be accepted by You.
Round 2
Reviewer 1 Report
The authors addressed properly the raised concerns, and the current version of the manuscript offers a clear and better description of the study hypothesis, questions, and goals.
Nevertheless, there is still the point of cost-related prescription non-redemption (crpnr), which must be better explained. By that, authors could address this “gap” in knowledge production, stressing the similarities (and differences) with out-of-pocket payment, proposing such work further research specification as seminal work on this topic.
Author Response
Dear Reviewer,
Thank you very much for the careful review of our manuscript. Your CRPNR related suggestion “Nevertheless, there is still the point of cost-related prescription non-redemption (crpnr), which must be better explained. By that, authors could address this “gap” in knowledge production, stressing the similarities (and differences) with out-of-pocket payment, proposing such work further research specification as seminal work on this topic.” has been considered. (Our working group deals a lot with prescription redemption associated problems and we use the CRPNR term daily.) The sentence which introduced the CRPNR term into the text has been reformulated/extended, to clarify the concept for readers not familiar with that. A reference has been inserted also to support the clarification.
Original version (last sentence in the second paragraph of the Introduction):
“Although it is known that most EU countries did not modify out-of-pocket payments for medication during the pandemic lockdown [9], the pandemic impact on cost related prescription redemption (CRPNR) has not yet been reported.”
Corrected version:
“Although it is known that most EU countries did not modify the copayment rules for medications (the proportion of costs paid by patients out of pocket) during the pandemic lockdown [9], the pandemic impact on the occurrence of patients’ inability to redeem medicine for financial reasons (cost related prescription redemption, CRPNR) has not yet been reported [24].”
The corresponding changes in the manuscript are shown with yellow highlighting in the manuscript.
Sincerely yours, Janos Sandor (on behalf of the authors)
Reviewer 3 Report
Dear Authors
Greetings
Congrats for the efforts, the work is better.
Kind regards
Author Response
Dear Reviewer, I would like to express our gratitude and thanks for your careful review! Your suggestions did improve our paper. Many thanks!
Authors